# Characterization of Colorants Formed by Non-Enzymatic Browning Reactions of Hydroxycinnamic Acid Derivatives

**DOI:** 10.3390/molecules27217564

**Published:** 2022-11-04

**Authors:** Leon Valentin Bork, Sascha Rohn, Clemens Kanzler

**Affiliations:** Institute of Food Technology and Food Chemistry, Department of Food Chemistry and Analysis, Technische Universität Berlin, Gustav-Meyer-Allee 25, 13355 Berlin, Germany

**Keywords:** non-enzymatic browning, hydroxycinnamic acids, phenolic compounds, 4-vinylcatechol, mass spectrometry, electrophilic aromatic substitution

## Abstract

The browning of plant-based food is commonly understood to result from the enzymatic polymerization of phenolic compounds to pigments, called melanin. However, during the thermal treatment of food, enzymes are deactivated, and non-enzymatic reactions predominate. The extent of the contribution of phenolic compounds to these non-enzymatic reactions has been speculated (“melanin-like vs. melanoidin-like”), but the literature is limited. Therefore, the aim of the present study was to investigate the heat-induced reactions of caffeic acid (CA), *para*-coumaric acid (CS), ferulic acid (FA), hydrocaffeic acid (HC), and 5-*O*-caffeoylquinic acid (CGA) under dry conditions. The model systems were characterized by color formation, reactant conversion, and antioxidant properties. Reaction products were analyzed by high-resolution mass spectrometry (HRMS) and nuclear magnetic resonance (NMR) spectroscopy. Decarboxylation could be classified as the driving force for the observed color formation and was significantly impacted by the substitution of the aromatic system. Reaction products were found to contribute to an increase in the antioxidant properties of the model systems. The oligomers described in this study could be incorporated into food melanoidins, contributing to the color and antioxidant properties observed in roasted food rich in phenolic compounds, such as coffee or cocoa.

## 1. Introduction

Thermal treatment is a vital step during the processing of most food items in modern diets. It promotes food safety by inactivating pathogens, degrading natural toxins, and by a conserving effect due to oxidative stabilization [1]. Additionally, heat-induced non-enzymatic reactions result in the chemical conversion of macronutrients, leading to the formation of flavor compounds [2], antioxidants [3], colorants [4], and macromolecules acting as dietary fibers [5]. However, undesired off-flavors [6] and process contaminants [7] are also described. Phenolic compounds have been proven to mitigate toxic effects by trapping reactive carbonyl intermediates [8,9] and, thereby, inhibiting the formation of heterocyclic aromatic amines and advanced glycation end-products (AGEs) [10]. Previous studies have revealed the high reactivity of phenolic compounds during thermic treatment, but little is known about their part in the Maillard reaction, especially regarding the formation of colorants [11,12,13]. α-Dicarbonyl compounds formed from the Maillard reaction such as 3-deoxyglucosone, methylglyoxal, and glyoxal were characterized as important reactants in browning reactions [14,15,16,17] and are prevalent in a large number of plant-based foods and ingredients [18]. So far, reactions of these Maillard reaction products and phenolic compounds described in the literature are limited to carbonyl trapping—aromatic, electrophilic substitution reactions between phenols and carbonyl groups—but these are primarily understood to inhibit non-enzymatic browning reactions [19,20,21]. Because of their two electrophilic carbonyl groups, reactive α-dicarbonyl compounds were found to exhibit the cross-linking of proteins via nucleophilic moieties such as amino groups [22]. However, the cross-linking of nucleophilic compounds by bifunctional carbonyls might be adaptable to phenolic compounds, uniting the concept of carbonyl trapping and the formation of high molecular weight colorants. This is relevant because, apart from the inhibitory effect on the Maillard reaction, phenolic compounds were also found to promote browning in other studies [23] but without a solid explanation for this observation. In addition, it is known that phenolic compounds are incorporated into food melanoidins in plant-based goods, most prominently coffee [24,25,26,27].

However, before the complex reactions in food can be understood, a detailed characterization of heat-induced non-enzymatic reactions exhibited by phenolic compounds is needed. In this context, the phenolic ingredients of coffee, most prominently chlorogenic acids and their corresponding hydrolysis products, caffeic acid (CA), ferulic acid (FA), or *para*-coumaric acid (CS), are suitable components for model studies. These account for up to 14% of the dry weight of green coffee beans and roasting was described to account for a significant loss of up to 95% of phenolic compounds [28]. Aside from thermal degradation reactions, the phenolic compounds are integrated into coffee melanoidins [26], but the underlying mechanism for the formation of phenol-containing melanoidins is unknown [29]. These data are needed to widen the knowledge of the impact on human nutrition, because coffee has been characterized as a major dietary source of melanoidins, contributing to an intake of up to 2 g per day [30].

The non-enzymatic oxidative browning of caffeic acid was investigated in the past, but the reaction conditions using a 100% oxygen atmosphere were not applicable to the roasting of food [31]. The solvent-free thermic-treatment of hydroxycinnamic acids was found to induce the decarboxylation and oligomerization of the respective decarboxylation products [11]. The formation of tetraoxygenated phenylindan isomers was first described by Stadler et al. [12] after the pyrolysis of caffeic acid at 230 °C. In a more recent study, Frank et al. [32] elucidated the structure of different di- and trimers formed by 4-vinylcatechol, the decarboxylation product of caffeic acid. These products were characterized as bitter-tasting substances and their formation could be demonstrated during coffee roasting [33]. However, there are still no data regarding their contribution to food color and melanoidin formation. The aim of the present study was to investigate the reactivity of structurally related phenolic compounds during roasting to widen the knowledge of substitution-dependent oligomerization, color formation, and changes in antioxidant properties.

Model experiments were carried out under dry roasting conditions by individually heating caffeic acid, 5-*O*-caffeoylquinic acid (CGA), *para*-coumaric acid, ferulic acid, and hydrocaffeic acid (HC) at 220 °C. Color formation was analyzed by *Vis* spectroscopy and the conversion of the reactants was measured by high-performance liquid chromatography coupled with a diode array detector (HPLC-DAD). The Trolox equivalent antioxidant capacity (TEAC) assay was used to determine the antioxidant capacity. Structural characterization was performed by high-resolution multiple-stage mass spectrometry (HRMS^n^), and the elucidation of 4-vinylcatechol dimers was achieved by one- and two-dimensional NMR experiments. The isolated reaction products were also characterized regarding their antioxidant capacity and reactivity by further heat treatment, individually or in combination with caffeic acid.

## 2. Results and Discussion

### 2.1. Browning Potential of Hydroxycinnamic Acid Derivatives under Roasting Conditions

To clarify the relevance of non-enzymatic browning reactions during the roasting of phenolic compounds, structurally related hydroxycinnamic acids—more precisely, caffeic acid, *para*-coumaric acid, ferulic acid, hydrocaffeic acid, and 5-*O*-caffeoylquinic acid as the main source of caffeic acid in coffee—were heated at 220 °C for up to 10 min. An analysis of the absorbance at 420 nm as an indication for the formation of brown colorants during the thermal treatment of food and model systems has been established as a useful method to characterize the reactivity of the investigated systems. Instead of absolute absorbance at 420 nm, the color index (Figure 1A), defined as absorbance at 420 nm per millimole of used reactant [14], is discussed. This allows a better comparison between model systems containing different amounts of a substance. Additional information on the colorants was obtained by measurements of the lightness (L*, Figure 1B), red/green intensity (a*, Figure 1C), and yellow/blue intensity (b*, Figure 1D) of each sample.

During the first 2.5 min of heating at 220 °C, the color intensity was highest for 5-*O*-caffeoylquinic acid (blue line, Figure 1A), but only slightly increased compared to caffeic acid (red line, Figure 1A). This might be a consequence of the starting conditions: The substance was obtained as a greenish powder leading to significant colored extracts at 0 min, whereas all the other phenol compounds used in this study were colorless. After 5 min, the color index was highest for caffeic acid (red line, Figure 1A), which showed a nearly linear increase in the color index throughout the incubation time of 10 min. After 10 min, the color index of caffeic acid was two-fold compared to 5-*O*-caffeoylquinic acid. The color formation of hydrocaffeic acid (green line, Figure 1A), ferulic acid (purple line, Figure 1A), and *para*-coumaric acid (brown line, Figure 1A) were comparatively low, resulting in color indices of around 10% (hydrocaffeic acid) and 3% (*para*-coumaric acid, ferulic acid) with regard to the values obtained for caffeic acid. An indication for the underlying mechanism of color formation could be drawn from a study by Rizzi and Boekley [11]: The authors described the decarboxylation reactions of caffeic acid, *para*-coumaric acid, and ferulic acid after pyrolysis at 205 °C for 45 min, resulting in a nearly complete conversion. The decarboxylation products were found to form dimeric compounds that were described as pale substances depending on their substitution. The 4-vinylcatechol dimer, deriving from caffeic acid, was reported as a pale-yellow oil, while dimers of 4-vinylguaiacol and 4-vinylphenol, formed after the decarboxylation of ferulic acid and *para*-coumaric acid, were colorless [11]. Nevertheless, consecutive reactions leading to further enlargement of the conjugated electron system must occur to explain the intense color formation observed in this study. In a recent investigation, it was reported that the electron-donating and -withdrawing groups of 4-hydroxycinnamic acids increase the decarboxylation rate during thermal treatment [13]. In combination with the data discussed herein, the hypothesis could be formulated that the substitution of the structurally related phenols significantly impacts their color formation under roasting conditions, whereby a higher electron density of the aromatic system correlates with higher extinction coefficients of the resulting reaction products.

As expected, the lightness decreased with an increase in the color index (Figure 1B). To be precise, the L* value significantly declined during the heat treatment of caffeic acid from almost 100 at 0 min (light grey bar) to 64 at 5 min (dark green bar) and to 38 at 10 min (dark grey bar). A comparable but less intense browning was exhibited for 5-*O*-caffeoylquinic acid, leading to a lightness of 59 at the end of the reaction. The roasting of hydrocaffeic acid (89), *para*-coumaric acid (95), and ferulic acid (94) only led to a small decrease in lightness. The analysis of the red (Figure 1C) and yellow (Figure 1D) color intensity of the model systems revealed that the yellow color intensity was overall more prevalent than the red color intensity, except for caffeic acid after 10 min of heating. Because the a* and b* values were always in the positive range, green or blue color tones were not observed in any system at any given time. The highest a* and b* values were obtained for caffeic acid: After 5 min, the yellow intensity was significantly higher than the red intensity, whereas both decreased over the course of further heating, resulting in a higher red content at 10 min. This decline could be explained by the decreasing lightness that impairs the determination of the underlying color intensities. The roasting of 5-*O*-caffeoylquinic acid led to a less sharp increase in a* and b* compared to caffeic acid, whereas the color intensity did not decline during the observed reaction time. For hydrocaffeic acid, *para*-coumaric acid, and ferulic acid, only a low increase in red content was detected after the roasting period of 10 min, but the increase in yellow intensity was significant, whereas the final b* values were comparable for these components. After 10 min, the yellow intensity was in the range between caffeic acid and 5-*O*-caffeoylquinic acid.

By analysis of the color indices, caffeic acid and 5-*O*-caffeoylquinic acid exhibited an elevated browning potential compared to *para*-coumaric acid, ferulic acid, and hydrocaffeic acid. The heat treatment of 5-*O*-caffeoylquinic acid was described to induce hydrolysis, yielding caffeic acid [32]. Consequently, the smaller increase in the color index of 5-*O*-caffeoylquinic acid compared to caffeic acid might be a result of a delayed release of caffeic acid and subsequent color formation. The color indices of hydrocaffeic acid, *para*-coumaric acid, and ferulic acid were low and negligible compared to caffeic acid and 5-*O*-caffeoylquinic acid. As there were significant differences regarding the L*, a*, and b* values between all investigated phenolic compounds, the assumption was made that roasting induced the formation of structurally different colorants at the observed reaction times. Overall, the color indices of the model systems correlated with the electron density and the size of the aromatic system of the reactants, resulting in the following ranking (from high to low): Caffeic acid > 5-*O*-caffeoylquinic acid > hydrocaffeic acid > ferulic acid > *para*-coumaric acid.

### 2.2. Thermally Induced Degradation of Hydroxycinnamic Acid Derivatives and Its Influence on pH

The conversion of the reactants during the reaction was analyzed by HPLC-DAD (Figure 2A). Earlier studies reported that the thermal treatment of these compounds induced decarboxylation [11,13,32]. In the present study, pH values were used as an indication of the prevalence of decarboxylation reactions (Figure 2B). The hypothesis was made that the degree of decarboxylation was reflected by an increase in the pH value of the methanolic extracts of the reaction mixtures because it results in the loss of the carboxyl moiety, which is mainly responsible for the p*K*_a_ values of the phenolic acids.

The conversion of the reactant was the highest and fastest in the caffeic acid model system. After 2.5 min, almost 98% of the initial caffeic acid was converted, and after 5 min, no caffeic acid was detectable. A slower but also almost complete conversion was obtained after the heating of 5-*O*-caffeoylquinic acid, with a relative conversion of 97% after 10 min. A turnover of 85% was determined for *para*-coumaric acid after 10 min, whereby its conversion was slow during the first half of the reaction and accelerated in the second half. In contrast, the reactivity of ferulic acid was comparatively low: After 5 min, 26% of the initially used amount was converted and further heat treatment did not result in a change in its concentration. Hydrocaffeic acid was even less reactive: 85% of the initial amount remained after 10 min of heating.

The pH values of the methanolic extracts of the samples were correlated inversely with the relative concentration of the reactants (Figure 2B): A sharp increase in pH value was measured after 2.5 min in the caffeic acid model system, which remained constant after the caffeic acid was completely converted (Figure 2A). A consistent but slower increase was obtained after the heating of 5-*O*-caffeoylquinic acid. The lower overall pH value compared to caffeic acid could be explained by the free carboxyl group of quinic acid, which has a lower p*K*_a_ value than the carboxyl group of caffeic acid [34]. The pH value of the *para*-coumaric acid model system did not significantly change during the first 5 min, but then sharply increased while its concentration declined. This indicated that the decarboxylation was delayed compared to caffeic acid but was the main mechanism for the observed conversion of *para*-coumaric acid. For ferulic acid, a small increase in pH was determined after 2.5 min but remained almost constant thereafter. Referring to the declining concentration during the first half of the reaction period, decarboxylation might not be the main mechanism for the degradation observed during the first 5 min. Instead, the pH value of the hydrocaffeic acid extract was constant in course of the reaction, indicating that its conversion was not a result of decarboxylation.

Overall, the data obtained by monitoring the conversion of the reactants and the pH value at given stages during the reaction showed that a decline in the concentration of the hydroxycinnamic acids was accompanied by an increase in the pH values of the methanolic extracts. With regard to earlier investigations discussing the decarboxylation of hydroxycinnamic acid derivatives [11,12,32], the correlation between reactant conversion and an increase in pH value observed in the present study could be interpreted as decarboxylation being the main reaction pathway for the conversion of caffeic acid, 5-*O*-caffeoylquinic acid, and *para*-coumaric acid. In contrast, the data obtained in the present study do not imply decarboxylation to be a key step during the roasting of ferulic and hydrocaffeic acid. Even though the decarboxylation of these phenolic compounds was investigated in earlier studies [12,32], this reaction mechanism was never discussed in the context of color formation. The data presented herein show that decarboxylation and subsequent oligomerization could possibly contribute to the generation of colorants within non-enzymatic browning reactions under roasting conditions.

### 2.3. Structural Characterization of Colored Oligomers Deriving from Hydroxycinnamic Acid Derivatives by High-Resolution Mass Spectrometry

High-resolution mass spectrometry was used to gain structural information about the reaction products formed during the dry heating of the phenolic compounds investigated. For analysis, ethyl acetate extracts obtained after roasting at 220 °C for 5 min were used, because the reaction was in an advanced stage, as apparent from the color indices and the relative concentration of the reactant. Ionization was carried out by electron spray ionization (ESI) in positive and negative ion mode as well as by atmospheric pressure chemical ionization (APCI) in positive ion mode. APCI resulted in a higher ion yield and a higher number of signals for reaction products formed after the treatment of caffeic acid, *para*-coumaric acid, and ferulic acid. Measurements in positive ion mode resulted in the detection of proton adducts, whereas other commonly formed sodium or potassium adducts were not found in significant intensities compared to the respective proton adducts. On the other hand, for hydrocaffeic acid, only ESI(−) resulted in usable HRMS spectra, whereas none of the applied methods resulted in the detection of reaction products formed after the treatment of 5-*O*-caffeoylquinic acid.

The APCI(+) scan spectrum of the extract resulting from the heat treatment of caffeic acid is shown in Figure 3. Based on their sum formulae, the most abundant signals were assigned to its decarboxylation product, 4-vinylcatechol. Tentative assignment can be found in Appendix A. Almost every signal highlighted in Figure 3 could be assigned to proton adducts of 4-vinylcatechol and further oligomerization products thereof. The base peak of *m/z* 137 was assigned to the proton adduct of vinylcatechol. *m/z* 269 and *m/z* 271 are representatives of vinylcatechol dimers with varying degrees of oxidation (–*n* × H_2_). This “redox pattern” of oligomerization products was also detected for vinylcatechol trimers with main signals at *m/z* 405 and 407, as well as its tetramers with *m/z* 541 and 543. *m/z* 163 was assigned to a fragment of a vinylcatechol dimer with a neutral loss of dihydroxybenzol (–C_6_H_6_O_2_). The fragmentation of phenolic compounds occurring as a consequence of APCI(+) has already been reported by other authors [35].

The HRMS analysis of equally treated solutions of *para*-coumaric acid (Appendix A), ferulic acid (Appendix A), and hydrocaffeic acid (Appendix A) revealed the formation of heterogeneous reaction products of the initial phenolic acid with its decarboxylated derivative. However, decarboxylation could only be observed for *para*-coumaric acid and ferulic acid in the form of proton adducts of 4-vinylphenol (*m/z* 121) and 4-vinylguaiacol (*m/z* 151). The HRMS-ESI(−) analysis of hydrocaffeic acid did not reveal the formation of the corresponding decarboxylation product 4-ethylcatechol at all. Instead, the pseudomolecular ions detected after the HRMS analysis of hydrocaffeic acid were predominantly assigned to condensation products of the native compound. In general, the incorporation of further phenolic acid molecules into oligomers could be explained by the formation of ester bonds by the aromatic hydroxy and aliphatic carboxyl groups. Overall, the abundance of pseudomolecular ions of the reactants with conserved carboxyl functions was significantly higher compared to decarboxylated monomers or heterogeneous oligomers of the respective reactants. These findings reveal that oligomerization products and, consequently, the underlying mechanism, differ depending on the substitution of the respective phenolic compound, whereby a higher degree of conjugation, as well as substituents stabilizing a negative charge, tend to promote decarboxylation and lead to higher color intensities of the resulting oligomeric product mixtures. Instead, for phenolic acids, other than caffeic acid, condensation reactions and ester formation were most likely the relevant pathway for oligomerization.

### 2.4. Isolation, Structural Elucidation, and Formation Mechanism of Colored 4-Vinylcatechol Dimers

By means of preparative HPLC, three different compounds from the reaction mixture of caffeic acid **1** were isolated and their structures were elucidated using one- and two-dimensional NMR spectroscopy. They were identified as dimers based on 4-vinylcatechol **2**; in detail, two stereoisomers of 1-(3,4-dihydroxyphenyl)-3-methyl-2,3-dihydro-1*H*-indene-5,6-diol **4** (P1 and P2) and (*E*)-4,4′-(but-1-ene-1,3-diyl)bis(benzene-1,2-diol) **3** (P3). These compounds, including different diastereomers, were already isolated by Frank et al. [32] after heating caffeic acid at 220 °C for 15 min. The relevance in the context of food processing was proven in a follow-up study by Blumberg et al. [33], who characterized these reaction products as bitter-tasting compounds that are formed during coffee roasting. The authors proposed that the protonation of 4-vinylcatechol to its corresponding cation was necessary prior to their dimerization, yielding **3** and **4** [12,32]. This mechanism was comparable to the proposal by Stadler et al. [12], who were the first researchers to describe the formation of two stereoisomers of **4**. However, solvent-free conditions, as applied in earlier studies [12,32] as well as in the present study, are unlikely to promote proton transfer to the extent that the reactants are fully converted after 2.5 min of heating. In a more recent investigation, it was reported that heat treatment in polar and protic solvents enabling proton transfer resulted in a yield twice as high compared to reactions performed in apolar media. In contrast to the data discussed herein, longer reaction times (30 min at 200 °C) were needed for a comparable conversion of caffeic acid, and no polymerization of 4-vinylcatechol was reported under these conditions [13]. In the present study, a revised mechanism that does not require any medium for proton transfer and makes the rapid conversion of caffeic acid more plausible is proposed in Figure 4: Under roasting conditions, caffeic acid **1** performs an intramolecular proton transfer and decarboxylation, yielding 4-vinylcatechol **2**. Subsequent reactions of 4-vinycatechol **2** with caffeic acid promote the decarboxylation and the formation of a 4-vinylcatechol dimer, 4,4′-(but-1-ene-1,3-diyl)bis(benzene-1,2-diol) **3**, in a concerted reaction via a six-membered transition state (Figure 3). The formation of the 1-(3,4-dihydroxyphenyl)-3-methyl-2,3-dihydro-1*H*-indene-5,6-diol **4** isomer results from a cyclization by the intramolecular electrophilic aromatic substitution of **3**. Subsequent reactions of **4** with caffeic acid **1**, via the proposed concerted mechanism or an electrophilic substitution reaction of **4** with 4-vinylcatechol, could explain the formation of larger oligomers, such as the trimer **5**, whose oxidized species were detected as *m/z* 405 and *m/z* 407 by HRMS. Both reactions are reasonable, whereas these might be strongly dependent on the concentration of the respective reactants. As 4-vinylcatechol catalyzes the decarboxylation of caffeic acid and subsequent oligomerization, the propagation of the reaction accelerates the conversion of caffeic acid, resulting in the fast conversion as observed by HPLC-DAD. This mechanism offers a plausible explanation for the accelerated decarboxylation of caffeic acid under dry conditions and the subsequent color formation on the basis of concerted oligomerization reactions.

### 2.5. Composition of Novel Colorants by High-Resolution Multiple-Stage Mass Spectrometry

Further analysis of oligomers found in the reaction mixtures of different hydroxycinnamic acids obtained after roasting was performed by HRMS^n^ experiments. As mentioned before, products of up to four units of the respective acids or their decarboxylation products could be found. For caffeic acid, most of the fragmentation pathways observed could be explained by the neutral loss of vinylcatechol (–C_8_H_8_O_2_) or its dimer (–C_16_H_14_O_4_). In addition, the loss of dihydroxybenzol (–C_6_H_6_O_2_) and dehydration (–H_2_O) were found to yield stable pseudomolecular ions. Oxidation (–n × H_2_) was induced by fragmentation, as well. Fragmentation spectra and a table for the tentative assignment of the fragment ions resulting from the fragmentation of seven oligomers detected in the scan HRMS spectrum can be found in the Appendix A.

A step-by-step example of the fragmentation of a pseudomolecular ion *m/z* 543 to its monomer *m/z* 137 is shown in Figure 5. Based on the sum formula C_32_H_30_O_8_, the tentative structure was composed of four vinylcatechol units that underwent one oxidation step.

The collision-induced dissociation (CID) of *m/z* 543 caused a neutral loss of vinylcatechol (–C_8_H_8_O_2_) and led to a trimer (*m/z* 407). Similarly, the loss of two vinylcatechol units (–C_16_H_14_O_4_) resulted in the fragment ion with *m/z* 271. Analogous fragmentation mechanisms were proposed for the fragment ion *m/z* 407: A loss of vinylcatechol (–C_8_H_8_O_2_) resulted in the detection of *m/z* 271, whereas the formal loss of the dimer (–C_16_H_14_O_4_) yielded the fragment ion of *m/z* 137. Further fragmentation of *m/z* 271 led to *m/z* 161 (–C_6_H_6_O_2_), *m/z* 143 (–C_6_H_6_O_2_, –H_2_O), and *m/z* 123 (–C_9_H_8_O_2_). In summary, the observed fragmentation reactions resulting in vinylcatechol as well as its dimer and trimer clearly showed that the pseudomolecular ion indeed comprises four units of vinylcatechol.

The data resulting from the fragmentation experiments of pseudomolecular ions *m/z* 285 and *m/z* 405 detected after the HRMS analysis of *para*-coumaric acid can be found in the Appendix A. The parent ion *m/z* 405 was composed of two units of vinylphenol and one unit of *para*-coumaric acid, and *m*/*z* 285 was composed of one unit each of *para*-coumaric acid and vinylphenol. These reaction products differed in their composition compared to caffeic acid, because aside from the decarboxylation product, the native phenolic compound was also incorporated into the dimer or the trimer. The most abundant fragment ions resulted from the neutral losses of dihydroxybenzol (–C_6_H_6_O_2_), water (–H_2_O), *para*-coumaric acid (–C_9_H_8_O_3_), and combinations thereof. Consequently, roasting induced the formation of a number of heterogeneous oligomers composed of *para*-coumaric acid and its decarboxylation product, vinylphenol.

Fragmentation spectra and a tentative assignment resulting from the tandem HRMS experiments of hydrocaffeic acid can be found in the Appendix A. These were performed for pseudomolecular ions formally assigned to an addition (*m/z* 363) and a condensation product (*m/z* 345) of two hydrocaffeic acid units. The fragmentation was characterized by neutral losses of hydrocaffeic acid (–C_9_H_10_O_4_) and water (–H_2_O). Thereby, the oligomerization of hydrocaffeic acid could be explained as simple addition or condensation reactions, but decarboxylation was only detected to a low degree, even after CID. Because the structure of hydrocaffeic acid does not provide an electrophilic center, a simple addition reaction of two hydrocaffeic acid molecules is unrealistic. A more reasonable explanation for the composition of the detected addition product is the formation of a water adduct of the respective condensation product during ionization. The formation of such water adducts has already been reported using the same conditions in previous investigations [14,15].

Due to the low abundance of pseudomolecular ions in the corresponding HRMS spectra of ferulic acid, CID experiments could not be performed for this hydroxycinnamic acid.

Results of the HRMS^n^ experiments strengthened the hypothesis that the substituents and the extent of the conjugated double-bond system significantly impacted decarboxylation reactions and the subsequent oligomerization of caffeic acid and structurally related phenols. The data clearly showed that decarboxylation is the driving force for the formation of oligomers with increased molecular weight for caffeic acid, whereas the analysis of *para*-coumaric acid revealed the formation of heterogeneous oligomers, including the reactants with conserved carboxylic acid functions, as well as the respective decarboxylated phenols. The decarboxylation of hydrocaffeic acid was less prevalent, and, instead, possibly addition but more likely condensation reactions of the native phenol could explain the composition of the predominant reaction products.

### 2.6. Impact of Heat-Induced Oligomerization on the Antioxidant Capacity

Although the chemical structures of selected lower molecular weight reaction products resulting from the roasting of 5-*O*-caffeoylquinic acid [32], caffeic acid [12], *para*-coumaric acid, and ferulic acid [11] have already been described in the literature, there are no data regarding their antioxidant properties. Consequently, the antioxidant capacity compared to Trolox was determined for the reaction mixtures of the phenolic compounds after heating times of 0, 5, and 10 min (Figure 6A), as well as for the isolated reaction products P1, P2, and P3 (Figure 6B).

The antioxidant capacity of the five native hydroxycinnamic acids was comparable. An increase in the antioxidant activity of the resulting reaction mixtures of caffeic acid and 5-*O*-caffeoylquinic acid was observed after thermal treatment at 220 °C for 5 and 10 min, whereas a significant decline could be determined for hydrocaffeic acid. A comparable but not statistically significant decline was observed after the heat treatment of ferulic acid. The antioxidant capacity of *para*-coumaric acid decreased in the first half of the reaction period, but heating for 10 min resulted in an increase in the antioxidant activity that was comparable to the unheated sample (0 min).

In addition to the antioxidant activity of the reaction mixtures, the antioxidant capacity of the isolated products P1, P2, and P3 was determined. Generally, the capacities of these products were significantly higher than the capacity of caffeic acid (Figure 6B): For P3, it was around two times higher and for P1 and P2 it was three times as high. These differences in antioxidant capacity could be explained by the different amount of oxidation steps enabled by the structure of the respective products. Native caffeic acid contains a single *ortho*-dihydroxybenzol moiety that might be oxidized once, yielding an *ortho*-chinone structure (Figure 6C). In contrast, all three dimers P1, P2, and P3 contain two *ortho*-dihydroxybenzol moieties. For P3, these are separated by an alkenyl bridge and each *ortho*-dihydroxybenzol moiety might undergo one oxidation step, resulting in two oxidations and consequently twice the antioxidant capacity as native caffeic acid. The oxidation of P1 and P2 results in the conjugation of both *ortho*-dihydroxybenzol moieties and, thereby, in an expansion of the aromatic system, enabling an additional oxidation step compared to P3. In consequence, the antioxidant capacity of P1 and P2 could have been expected to be around three times higher compared to caffeic acid. Despite their different absolute configuration, the antioxidant properties of P1 and P2 were equal.

The increase in the antioxidant activities of the reaction mixtures observed during the roasting of caffeic acid and 5-*O*-caffeoylquinic could be explained by the formation of vinylcatechol dimers, such as P1, P2, and P3. Because the antioxidant capacities of these reaction products were significantly higher compared to the antioxidant activity of the reaction mixtures obtained after 5 and 10 min of the thermal treatment of caffeic acid, the hypothesis could be formulated that further oligomerization does not result in a linear increase in the antioxidant capacity relative to caffeic acid. This hypothesis is strengthened by findings published by Plumb et al. [36], who studied the antioxidative properties of epicatechin and its di-, tri-, and tetramer. The dimer, procyanidin B2, exhibited a two-fold antioxidative capacity compared to epicatechin, but only a slight increase compared to the dimer was determined for the trimer. The addition of another epicatechin unit resulted in a decrease in the antioxidant capacity.

The heat treatment of *para*-coumaric acid resulted in a decline in the antioxidant activity in the first 5 min of the reaction. Further heat treatment was accompanied by a significant increase to an elevated value after 10 min. Because *para*-coumaric acid is missing one electron-donating hydroxy group compared to caffeic acid, the rate of the decarboxylation of *para*-coumaric acid is slower and the formation of structurally related oligomers exhibiting increased antioxidant capacity is delayed.

In contrast, the oligomerization of hydrocaffeic acid and ferulic acid was proposed to result from condensation reactions, as discussed before. The condensation of the carboxyl function with an aromatic hydroxy group would inhibit the antioxidant activity of the *ortho*-dihydroxy moiety, resulting in a decrease in the antioxidant capacity.

### 2.7. Investigation of the Polymerization Mechanism by High-Resolution Mass Spectrometry

Further information on the polymerization of the reaction products P1, P2, and P3 was obtained by HRMS spectra and HRMS^n^ experiments of the native substances and the reaction mixtures obtained after heating (5 min, 220 °C) with or without the addition of caffeic acid. The aim of these experiments was to investigate if the dimers could further polymerize on their own and if caffeic acid or its decarboxylation products would promote polymerization. The corresponding scan spectra are shown in Figure 7A–F. The tentative assignments of the main signals are found as Appendix A. The equivalent treatment of P2 provided comparable results to P1 and can also be found in the Appendix A.

The APCI(+)-induced in-source fragmentation of the vinylcatechol dimers P1 (Figure 7A) and P3 (Figure 7B) is reflected by a neutral loss of dihydroxybenzol (–C_6_H_6_O_2_) and the detection of *m/z* 163 as a base peak. Further, the analysis of P1 resulted in the sole detection of the oxidized species *m/z* 271 (–H_2_), whereas the redox pairs of vinylcatechol trimers and tetramers were detected for P3 with low intensities. The exact chemical structure and number of carbon and hydrogen atoms were determined by NMR spectroscopy (Appendix A). The occurrence of higher oligomers, especially for P3 (Figure 7B), could be a consequence of adduct formation during ionization or impurities of the sample.

After the heat treatment of P1 and P3, *m/z* 271 was the base peak. This could be a result of ring-opening and/or cyclization to thermodynamically preferred isomers that are less prone to undergo fragmentation under APCI(+) conditions. For both P1 (Figure 7C) and P3 (Figure 7D), redox pairs of vinylcatechol trimers (P1: *m/z* 401, *m/z* 403, *m/z* 405, *m/z* 407; P3: *m/z* 401, *m/z* 403, *m/z* 405, *m/z* 407, *m/z* 409) and its tetramers (P1: *m/z* 535, *m/z* 537, *m/z* 539, *m/z* 541; P3: *m/z* 535, *m/z* 537, *m/z* 539, *m/z* 541, *m/z* 543, *m/z* 545) were found in a high abundance, which was even significantly higher for P3 compared to P1. This indicated that both dimers were able to undergo subsequent reactions resulting in larger structures. The fact that, besides tetramers, trimers were also formed from the dimers, showed that the latter also undergo cleavage reactions resulting in vinylcatechol, which then reacted with other dimers. Pseudomolecular ions *m/z* 409 and *m/z* 545 exclusively detected after the heat treatment and analysis of P3 could be assigned to oligomers of P3 and vinylcatechol, whereas products assigned to P1 underwent at least one oxidation step. In contrast, the impact on the formation of tri- and tetramers after roasting P2 was negligible (Appendix A). The higher abundance of oxidized species could be explained by the different structural properties of P1 and P3: The cyclic indane structure exceeds the higher antioxidant activity compared to the aliphatic-linked *ortho*-dihydroxybenzol moieties, as present in P3. Further, one oxidation step of the indane P1 induces aromatization, yielding an aromatic indene, which exhibits a higher stabilization by resonance.

The presence of caffeic acid during the incubation of P1 and P3 significantly increased the abundance of discussed trimeric and tetrameric species: For P1, additional signals could be assigned to a trimer (*m/z* 401) and a tetramer (*m/z* 539) (Figure 7E). Based on the relative intensity of the respective signals, the formation of the trimer *m/z* 405 was favored for P3 (Figure 7F), which was the base peak after heat treatment. The relative abundance of tetramers did not change compared to samples of P1 and P3 heated without the addition of caffeic acid, whereas an increase in the relative intensity was observed after incubating P2 with caffeic acid. These observations support the proposed reaction mechanism shown in Figure 4: The concerted decarboxylation and the addition of the vinylcatechol body. Hence, the decarboxylation of caffeic acid is the driving force of this reaction, and the formation of higher oligomers by the reaction of vinylcatechol dimers or other oligomers is less relevant.

Overall, these experiments indicate that the oligomerization of vinylcatechol and its dimers depends on the constitution. As P1 and P3 were found to favor the formation of a trimer after the addition of caffeic acid, an increase in the trimer and tetramer resulted from the incubation of P2. The composition of various signals was verified by multi-stage HRMS experiments. Independently of the reactant, the composition could be assigned to different redox stages and oligomers of vinylcatechol, whereas the exact constitution could not be determined. The fragmentation spectra and tentative assignments can be found in the Appendix A (P1: Appendix A; P2: Appendix A; P3: Appendix A).

## 3. Materials and Methods

### 3.1. Chemicals

Acetic acid was purchased from Carl Roth GmbH + Co., KG (Karlsruhe, Germany). Acetonitrile, ethyl acetate, and methanol were purchased from VWR International GmbH (Darmstadt, Germany). Acetonitrile-d_3_, 2,2′-azinobis(3-ethylbenzothiazoline-6-suflonate), caffeic acid, chlorogenic acid, and potassium persulfate were purchased from Sigma-Aldrich Chemie GmbH (Steinheim, Germany). Ferulic acid, hydrocaffeic acid, and *para*-coumaric acid were purchased from Fluka Chemicals Ltd. (Wales, UK). Potassium dihydrogen phosphate and potassium hydrogen phosphate were purchased from Merck KGaA (Darmstadt, Germany).

### 3.2. Incubation of Phenolic Compounds under Roasting Conditions

Model roasting was performed by incubating 0.05 mmol of a phenolic compound. Caffeic acid (9.0 mg), 5-*O*-Caffeoylquinic acid (17.7 mg), *para*-coumaric acid (8.2 mg), ferulic acid (9.7 mg), and hydrocaffeic acid (9.1 mg) were heated individually in sealed glass vials at 220 °C. At defined reaction times (0, 2.5, 5.0, 7.5, and 10.0 min), samples were taken and cooled down in a freezer to −20 °C to stop the reaction. For color and pH measurements, HPLC analysis, and TEAC, the residue was taken up in 1.0 mL of methanol. Every sample was prepared in triplicate (all results are given as ± standard deviation). Measurements of pH were performed using a Mettler Toledo Five Easy™ pH meter (Mettler-Toledo GmbH, Gießen, Germany). To avoid the detection of methylated derivatives, the residues of equally treated samples were taken up in ethyl acetate (for caffeic acid, *para*-coumaric acid, ferulic acid, and hydrocaffeic acid) or water/acetonitrile (50/50, *v*/*v*) (for chlorogenic acid) prior to HRMS analysis.

### 3.3. Color Measurements

The color of the methanol extracts was characterized by L* a* b measurements as defined by The International Commission on Illumination (CIE) [37]. Visible spectra (400–800 nm) of the samples were recorded with a spectrophotometer (Specord 200 Plus, Analytik Jena GmbH, Jena, Germany; software: WinAspect Plus Version 4.3). For illumination, a D65 lamp with an angle of 2° was used. The observer angle was set to 10°.

For the characterization of the brown color, additional measurements at 420 nm were conducted. Samples were diluted in methanol when the extinction exceeded 1.0. To allow a comparison of all model systems, the results of the measurements at 420 nm are given as color index, which was defined as the absorbance at 420 nm multiplied by the dilution factor per millimole of the reactants (A_420_ × F)/n_i_.

All measurements were performed in a quartz cuvette against methanol.

### 3.4. HPLC-DAD Analysis of Phenolic Compounds

Caffeic acid, chlorogenic acid, *para*-coumaric acid, ferulic acid, and hydrocaffeic acid were identified by HPLC with reference standards and quantified relative to samples without treatment (t = 0 min) by HPLC-DAD. For separation, a Shimadzu analytical HPLC system (Shimadzu Corp, Kyōto, Japan) with the following setup was used: pump, Shimadzu LC-9A; degasser, DG-1300 (Knauer Wissenschaftliche Geräte GmbH, Berlin, Deutschland); autosampler, Shimadzu SCL-6B and SIL-6B; column, Prodigy™ ODS-3 C18 (Phenomenex Ltd. Deutschland, Aschaffenburg, Deutschland); detector, Shimadzu SPD-M10A; and software, Shimadzu Class-LC10 v1.64A. The following settings were used: column temperature, 35 °C; flow rate, 0.5 mL/min; eluent A, water with 0.5 vol.% acetic acid; eluent B, acetonitrile; eluent gradient, 0 min, 97.5% A; 5 min, 10% A; and 11 min; 97.5% A; and wavelength for quantitation, 270 nm (hydrocaffeic acid) and 310 nm (caffeic acid, chlorogenic acid, *para*-coumaric acid, and ferulic acid).

### 3.5. Isolation of 1-(3,4-Dihydroxyphenyl)-3-methyl-2,3-dihydro-1H-indene-5,6-diol (P1, P2) and (E)-4,4′-(But-1-ene-1,3-diyl)bis(benzene-1,2-diol) (P3)

Caffeic acid (90 mg, 0.5 mmol) was heated at 220 °C in a sealed glass vial for 5 min. To stop the reaction, the sample was cooled down in a freezer to −20 °C. The residue was taken up in 5 mL acetonitrile/water (3:7, *v*/*v*) and the extract was purified by preparative HPLC. For separation, an Agilent 1200 Series preparative HPLC system (Agilent Technologies, Waldbronn, Germany) with the following setup was used: two preparative pumps G1361A; dual loop autosampler G2258A; column, NUCLEODUR 100-5 C18 ec (MACHEREY-NAGEL GmbH and Co., KG, Düren, Germany); column oven, Merck-Hitachi T-6300 (Merck KGaA, Darmstadt, Germany); detector, MWD G1365B; sample collector, prep FC G1364B; and software, Agilent ChemStation B.02.01-SR1.

The following settings were used: column temperature, 35 °C; flow rate, 10 mL/min; eluent A, water; eluent B, acetonitrile; eluent gradient, 0 min, 30% A; 27 min, 40.1% A; 29 min, 85% A; and 35 min, 85% A; and wavelength for identification, 285 nm. Fractions were collected based on retention time. Two diastereomers of 1-(3,4-dihydroxyphenyl)-3-methyl-2,3dihydro-1*H*-indene-5,6-diol were collected from 22.35 min to 22.85 min (P1, oily-red) and 23.15 min to 23.70 min (P2, oily-red). (*E*)-4,4′-(but-1-ene-1,3-diyl)bis(benzene-1,2-diol) (P3, oily-brown) was collected between 24.00 and 24.60 min. This procedure was repeated until sufficient material was isolated to allow structural elucidation by NMR and HRMS analysis. Spectroscopic data are in line with Frank et al. [32].


*Product P1:*


^1^H NMR {400 MHz, acetonitrile-d_3_}: δ_H_ 6.69 (d, 1H, CH, 4), 6.68 (d, 1H, CH, 5′), 6.52 (d, 1H, CH, 2′), 6.44 (m, 1H, CH, 6′), 6.39 (s, 1H, CH, 7), 4.15 (t, 1H, CH, 1), 3.18 (m, 1H, CH, 3), 2.13 (m, 1H, CH, 2), 2.03 (m, 1H, CH, 2), 1.16 (d, 3H, CH_3_, 3″).

^13^C NMR (100 MHz, acetonitrile-d_3_): δ_C_ 145.3 (s, C3′),144.6 (s, C6), 144.2 (s, C5), 143.6 (s, C4′), 141.6 (s, C7a), 139.7 (s, C1′), 138.9 (s, C3a), 120.2 (s, C6′), 116.1 (s, C5′), 115.4 (s, C2′), 112.3 (s, C7), 111.0 (s, C4), 49.3 (s, C1), 45.8 (s, C2), 38.5 (s, C3), 21.1 (s, C3″). Oxidized species detected by HRMS (APCI+). Calculated for C_16_H_14_O_4_H^+^, 271.0967; found, 271.0961. Chemical structure and assignments are shown in Appendix A.


*Product P2:*


^1^H NMR {700 MHz, acetonitrile-d_3_}: δ_H_ 6.74 (d, 1H, CH, 5′), 6.67 (s, 1H, CH, 4), 6.62 (d, 1H, CH, 2′), 6.56 (dd, 1H, CH, 2′), 6.24 (s, 1H, CH, 7), 3.95 (q, 1H, CH, 1), 3.00 (m, 1H, CH, 3), 2.56 (m, 1H, CH, 2), 1.42 (m, 1H, CH, 2), 1.25 (d, 3H, CH_3_, 3″).

^13^C NMR (175 MHz, acetonitrile-d_3_): δ_C_ 145.3 (s, C3′), 144.5 (s, C6), 144.0 (s, C4′), 143.8 (s, C5), 141.4 (s, C7a), 139.6 (s, C1′), 138.7 (s, C3a), 120.8 (s, C6′), 116.1 (s, C5′), 115.9 (s, C2′), 112.1 (s, C7), 110.6 (s, C4), 50.2 (s, C1), 47.4 (s, C2), 38.6 (s, C3), 20.0 (s, C3″). Oxidized species detected by HRMS (APCI+). Calculated for C_16_H_14_O_4_H^+^, 271.0967; found, 271.0965. Chemical structure and assignments are shown in Appendix A.


*Product P3:*


^1^H NMR {700 MHz, acetonitrile-d_3_}: δ_H_ 6.84 (s, 1H, CH, 3′), 6.73 (s, 1H, CH, 6), 6.72 (m, 1H, CH, 6′), 6.71 (m, 1H, CH, 3), 6.71 (m, 1H, CH, 5′), 6.61 (dd, 1H, CH, 1), 6.24 (d, 1H, CH, 1″), 6.14 (m, 1H, CH, 2″), 3.43 (t, 1H, CH, 3″), 1.32 (d, 3H, CH_3_, 4″).

^13^C NMR (175 MHz, acetonitrile-d_3_): δ_C_ 145.6 (s, C2), 145.4 (s, C2′), 144.9 (s, C1′), 143.6 (s, C1), 139.5 (s, C4), 134.3 (s, C2″), 131.6 (s, C4′), 128.6 (s, C1″), 119.7 (s, C5), 119.6 (s, C5′), 116.3 (s, C6), 116.2 (s, C6′), 115.2 (s, C3), 113.6 (s, C3′), 42.7 (s, C3″), 21.9 (s, C4″). HRMS (APCI+) calculated for C_16_H_16_O_4_H^+^, 273.1119; found, 273.1121. Chemical structure and assignments are shown in Appendix A.

### 3.6. Incubation of Compounds P1, P2, and P3

1-(3,4-Dihydroxyphenyl)-3-methyl-2,3dihydro-1*H*-indene-5,6-diol (2.7 mg) and (*E*)-4,4′-(but-1-ene-1,3-diyl)bis(benzene-1,2-diol) (1.3 mg) were incubated individually as well as in equimolar mixtures with caffeic acid (P1/P2: 1.8 mg, P3: 0.9 mg) at 220 °C for 5 min. To stop the reaction, samples were cooled down in a freezer to −20 °C. The residue was extracted in ethyl acetate, diluted (1 mg/mL), and analyzed by APCI(+)-HRMS.

### 3.7. Trolox Equivalent Antioxidant Capacity Assay

Trolox equivalent antioxidant capacity (TEAC) assay was performed as described by Kanzler et al. [38]. An aqueous solution of 2,2′-azino-bis-(3-ethylbenzothiazoline-6-sulfonic acid) solution (ABTS; 10 mmol/L) was mixed with an aqueous solution of potassium persulfate (3.5 mmol/L). For the preparation of the radical cation solution, the mixture was incubated overnight at room temperature. The working radical solution was prepared by diluting the radical cation solution 12:100 with phosphate buffer (PBS) (5 mmol/L phosphate, pH 7.2–7.4). Calibration was performed with six Trolox standards (0.01, 0.02, 0.04, 0.06, 0.08, and 0.1 mmol/L; diluted in phosphate buffer). An amount of 500 µL of the working solution and 500 µL of the samples (diluted with phosphate buffer) were mixed. Extinction at 734 nm was measured using a Biotek Uvikon XL (Agilent Technologies Inc., Santa Clara, CA, USA) after an incubation time of 120 min. The extinction was multiplied with the dilution factor to allow a comparison of all samples.

### 3.8. APCI(+) and ESI-Orbitrap Multiple-Stage High-Resolution Mass Spectroscopy

HRMS^n^ analyses were carried out as described before [39]: A Thermo Fisher Scientific Inc. LTQ Orbitrap XL™ instrument equipped with an Ion Max™ Source (Waltham, MA, USA) was used. Measurement of the samples (1 mg/mL in acetonitrile/water (50/50, (*v*/*v*) or ethyl acetate) was performed in positive and negative ion mode (ESI) as well as in positive ion mode (APCI) by direct infusion. Reserpine (0.05 mg/mL) was used for mass calibration. The normalized collision energy of collision-induced dissociation varied from 5 to 50%. For the interpretation of the mass spectra, the software Freestyle 1.6 was used (Thermo Fisher Scientific Inc., Waltham, MA, USA).

### 3.9. NMR Spectroscopy

NMR spectra were recorded on either a Bruker Avance™ II 400 MHz (Bruker Corporation, Billerica, MA, USA) operating at 400 MHz (^1^H NMR) or 100 MHz (^13^C NMR) or a Bruker Avance™ III 700 MHz spectrometer operating at 700 MHz (^1^H NMR) or 175 MHz (^13^C NMR). Standard one-dimensional (^1^H, ^13^C, DEPT) and two-dimensional experiments (HSQC, HMBC, COSY) were employed for structural elucidation. NMR shifts were referenced to the residual signal of the solvent acetonitrile-d_3_ according to [40].

### 3.10. Statistical Analysis

All samples were prepared and analyzed in triplicate. All results are shown as means ± standard deviation. Significant differences (*p* < 0.05) were analyzed by one- or two-way analysis of variance (ANOVA) and Tukey’s test using the GraphPad Prism 8.0.2 software (San Diego, CA, USA).

## 4. Conclusions

In the present study, the reactivity of caffeic acid and structurally related phenolic acids was characterized with a focus on their contribution to non-enzymatic browning reactions. Incubation under roasting conditions revealed that polymerization and, thereby, the browning intensity of the reaction mixture strongly depends on electron density and the size of the aromatic ring system of the reactant. Decarboxylation reactions were identified as the driving force of the oligomerization of caffeic acid. In the process of decarboxylation, 4-vinylcatechol undergoes a concerted addition reaction with another vinylcatechol body or another vinylcatechol oligomer. Hence, the presence of reaction products deriving from caffeic acid accelerated the decarboxylation of caffeic acid and oligomerization by the in situ incorporation of 4-vinylcatechol.

In essence, this study provides crucial information on the contribution of phenolic compounds to the underlying mechanism of non-enzymatic color formation and the structural background of the increase in antioxidant properties during non-enzymatic browning reactions. The herein-discussed data are mostly relevant for plant-based goods containing caffeic acid, most prominently coffee [26]. Combining the concepts for the non-enzymatic formation of colorants proposed in this study and recent findings on the oligomerization of Maillard reaction intermediates [14,15,17] is helpful for the structural clarification of coffee melanoidins that are already known for their high content of phenolic compounds. In this context, electrophilic aromatic substitution, as already described for carbonyl trapping, should be considered a key mechanism for the linkage of phenol-deriving colorants and melanoidins via carbonyl moieties or unsaturated domains.

## Figures and Tables

**Figure 1 molecules-27-07564-f001:**
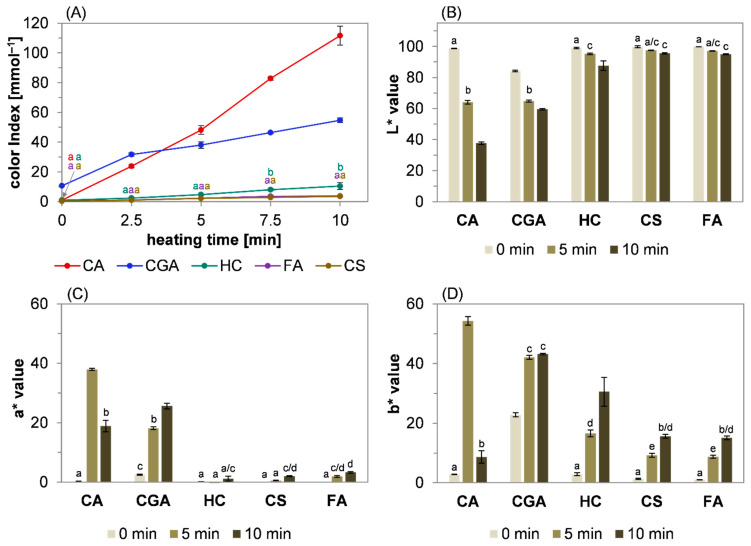
Time-dependent color formation of different hydroxycinnamic acid derivatives. Change in (**A**) the color index of caffeic acid (CA, red line), 5-*O*-caffeoylquinic acid (CGA, blue line), hydrocaffeic acid (HC, green line), *para*-coumaric acid (CS, brown line), and ferulic acid (FA, purple line) and (**B**) lightness (L*), (**C**) red/green intensity (a*), and (**D**) yellow/blue intensity (b*) over the course of roasting at 220 °C. Statistical analyses were performed by two-way ANOVA and Tukey’s test (*p* < 0.05). Statistically equal values of the data points are designated by equal letters.

**Figure 2 molecules-27-07564-f002:**
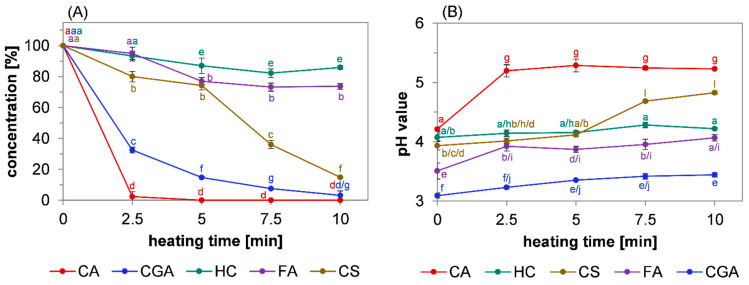
Reactivity of different hydroxycinnamic acid derivatives. Heat-induced (**A**) conversion of caffeic acid (CA, red line), 5-*O*-caffeoylquinic acid (CGA, blue line), hydrocaffeic acid (HC, green line), *para*-coumaric acid (CS, brown line), and ferulic acid (FA, purple line) over the course of roasting at 220 °C and (**B**) changes in the pH values of the methanolic extracts. Data were analyzed by two-way ANOVA and Tukey’s test (*p* < 0.05). Statistical differences between the data points are designated by different letters.

**Figure 3 molecules-27-07564-f003:**
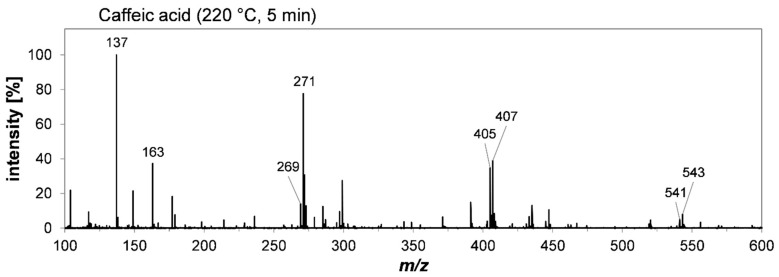
Non-enzymatic oligomerization of caffeic acid under roasting conditions (220 °C, 5 min). The high-resolution mass spectrum of the reaction mixture was modified by removing the signal of the solvent (dimer of ethyl acetate) and re-referencing the intensity of all signals accordingly. A scan spectrum of the solvent is attached and can be found in the Appendix A.

**Figure 4 molecules-27-07564-f004:**
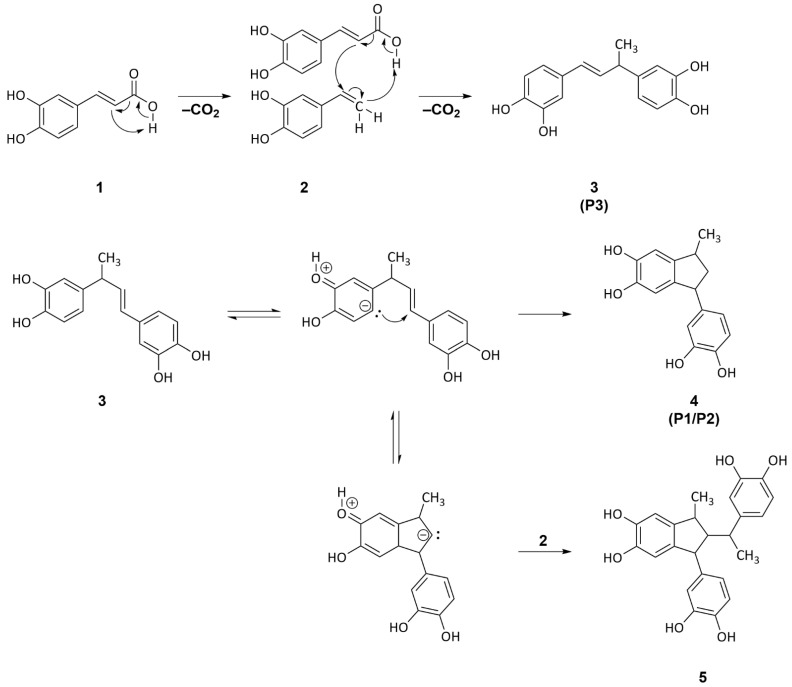
Decarboxylation of caffeic acid **1** and the subsequent oligomerization of 4-vinylcatechol **2** leads to the formation of (*E*)-4,4′-(but-1-ene-1,3-diyl)bis(benzene-1,2-diol) **3** (P3). Cyclization via an intramolecular electrophilic substitution reaction of **3** results in the formation of 1-(3,4-dihydroxyphenyl)-3-methyl-2,3dihydro-1*H*-indene-5,6-diol **4** (P1, P2), whereas the addition of 2 before restoring aromaticity leads to the trimer **5**.

**Figure 5 molecules-27-07564-f005:**
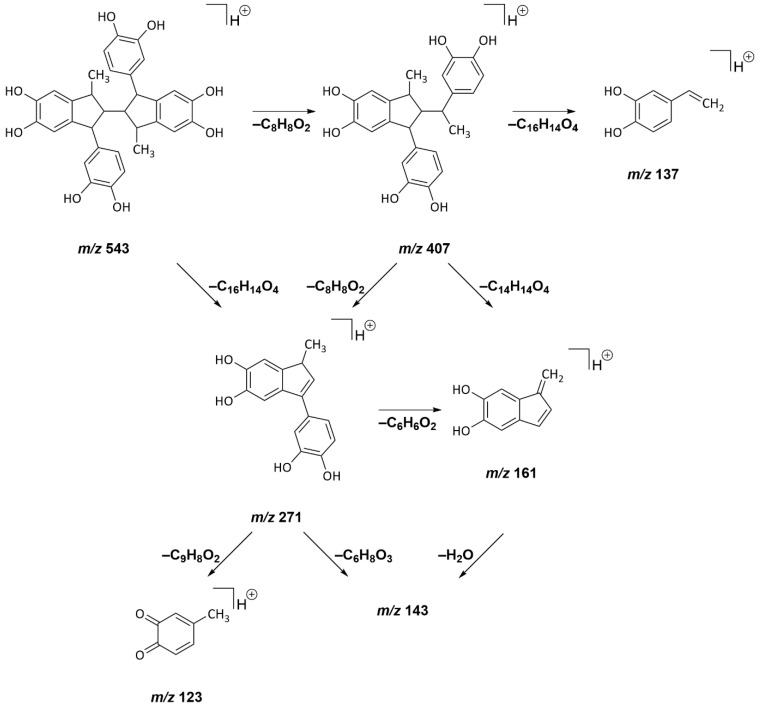
Fragmentation pathways for the pseudomolecular ion *m/z* 543, as analyzed by means of APCI(+)-HRMS^n^ experiments.

**Figure 6 molecules-27-07564-f006:**
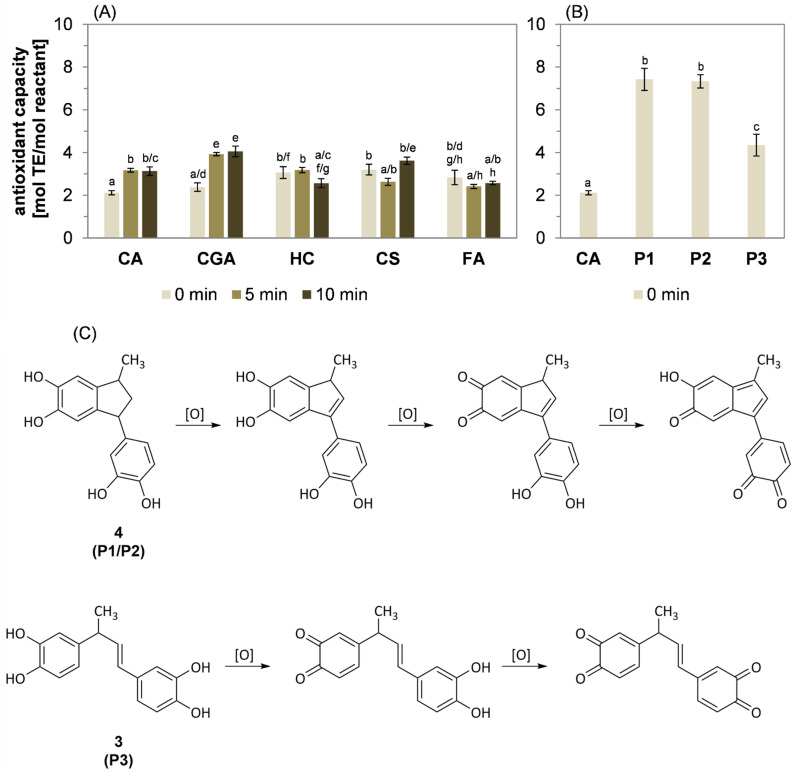
Antioxidant properties of phenolic compounds. (**A**) Antioxidant activity of reaction mixtures of caffeic acid, chlorogenic acid, hydrocaffeic acid, *para*-coumaric acid, and ferulic acid. (**B**) Antioxidant capacity of caffeic acid and reaction products **3** (P3) and **4** (P1/P2) at 0 min compared to Trolox, given as Trolox equivalents (TE). (**C**) Proposed oxidation reactions of products **3** and **4**. Statistically significant values (*p* < 0.05) were analyzed by (**A**) two-way ANOVA and Tukey’s test and (**B**) one-way ANOVA and Tukey’s test. Statistical differences within the columns are designated by different letters.

**Figure 7 molecules-27-07564-f007:**
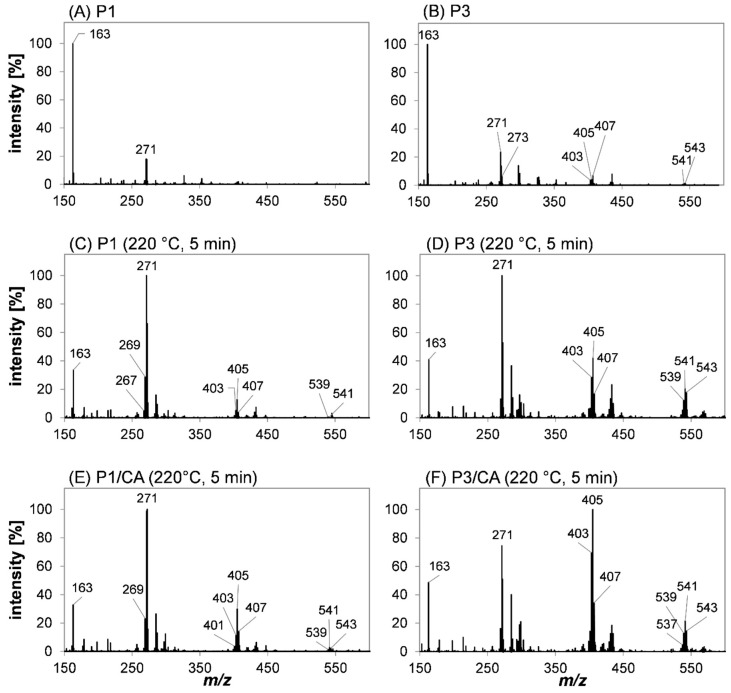
HRMS scan spectra of (**A**) P1 and (**B**) P3 before heat treatment, as well as (**C**) P1, (**D**) P3, (**E**) P1/CA, and (**F**) P3/CA after heat treatment for 5 min at 220 °C using an APCI-orbitrap-MS instrument in positive ion mode. Mass spectra were modified by removing the signal of the solvent (dimer of ethylacetate) and re-referencing the intensity of all signals accordingly. A scan spectrum of the solvent is attached and can be found in the Appendix A.

## Data Availability

The data presented in this study are available on request from the corresponding author.

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
