# Peer review of "Characterization of Colorants Formed by Non-Enzymatic Browning Reactions of Hydroxycinnamic Acid Derivatives"

_molecules, 2022, doi:10.3390/molecules27217564_

Round 1
Reviewer 1 Report
Estimated authors
It seems to me a fairly complete and interesting study of the reactivity of the phenolic compounds contained in coffee during its roasting, the reaction products and its antioxidant capacity. Find some details that I mention below (As well as in the included pdf)
1) L 93 and L 230. I suggest expressing “….Hydroxycinnamic Acids…” instead of “…Hydroxycinnamic Acid…”, since the term refers to the group of 5 phenolic acids studied
2) L 113, Correct “caffeoyquinic” to “caffeoylquinic”
3) L 226. Include bibliographical reference after “…earlier studies…”
4) L 247 and L 250. Reference is made to “…Figure 3A…” however Figure 3 does not contain part A or part B
5) L 315 Reference is made to “…Figure 3B…” however Figure 3 does not contain part A or part B
6) Figure 6A and and 6B. Specify the units of the y-axis: molTE/(mol of ….), specify the units of the denominator. Indicate in the description of this figure, the meaning of the lowercase letters a, b, c written on the columns
7) Line 631. After “…2,2′-azino-bis-(3-ethylbenzothiazoline-6-sulfonic acid)” write the abbreviation of the salt (ABTS)

Author Response
Please see the attachement.

Reviewer 2 Report
1. After the caffeic acid was treated at 220 ℃ for 5 minutes, the appearance of dimer and trimer was very interesting. However, these dimers and trimers are not necessarily directly related to the generation of the final melanin color. At least the relevant evidence provided in the manuscript is insufficient.
2.Furthermore, mass spectrometry is very important for the determination of the molecular formula, but it is difficult to distinguish the mass spectra of the mixtures. Therefore, the use of chromatography for separation before mass spectrometry analysis will increase the credibility of mass spectrometry. Please provide detailed chromatograms and mass spectrum combination pictures in the main text. In particular, please provides high resolution mass spectrum data.
3. ANOVA and post-hoc Tukey test needs to be used for variance analyses in the data in Figures 1 and 2.
4. According to the NMR spectra of the products, the chemical structures of the compounds 4 and 5, including their stereo configurations, should be identified carefully.
Author Response
Please see the attachement.

Round 2
Reviewer 2 Report
Colorants formed by non-enzymatic browning reactions of hydroxycinnamic acid derivatives is interested for investigation. The background irrelevant to this study in the abstract can be deleted appropriately. This manuscript has been improved in the statistical analyses and qualitative data analyses. In the results and discussion, especially in Figure 6, the change pathway of the products obtained in the experiment and pigment substances were deduced, which will increases the innovation and research significance of this manuscript. If the absolute configurations of P1-P3 can be determined, then the experiments will be more clear. I suggest that this manuscript can be accepted for publication at this version.
Author Response
Please see the attachement.
